



# Sea Spray Fluxes from the Southwest Coast of the United Kingdom –

# Dependence on Wind Speed and Wave Height

Mingxi Yang[1], Sarah J. Norris[2], Thomas G. Bell[1], Ian M. Brooks[2]

[1] Plymouth Marine Laboratory, Prospect Place, Plymouth, UK PL1 3DH.

[2] Institute for Climate and Atmospheric Science, School of Earth and Environment, University of Leeds, Leeds, UK

*Corresponding author: Mingxi Yang (miya@pml.ac.uk)*

**Abstract.** Fluxes of sea spray aerosols were measured with the eddy covariance technique from the Penlee Point Atmospheric Observatory (PPAO) on the southwest coast of the United Kingdom over several months from 2015 to 2017. Two different fast-responding aerosol instruments were employed: an ultra-fine condensation particle counter (CPC) that detects aerosols with radius above ca. 1.5 nm, and a compact lightweight aerosol spectrometer probe (CLASP) that provides a size distribution between ca. 0.1 and 6 μm. The aerosol fluxes were almost always from sea to air, indicating sea spray emission. Fluxes from

the CPC and from the CLASP (integrated over all sizes) were generally comparable, implying a reasonable closure in the aerosol number flux. Compared to most previous observations over the open ocean, at the same wind speed the mean sea spray number fluxes at PPAO are much greater. Significant wave height and wave Reynolds number explain more variability in sea spray fluxes than does wind speed, implying that enhanced wave breaking resulting from shoaling in shallow coastal waters is a dominant control on sea spray emission. Comparisons between two different wind sectors (open water vs. fetch-limited

Plymouth Sound) and between two sets of sea states (growing vs. falling seas) further confirm the importance of wave characteristics on sea spray fluxes. These results suggest that spatial variability in wave characteristics need to be taken into account in predictions of coastal sea spray productions and also aerosol loading.

## 1. Introduction

Sea spray aerosols formed from wave breaking impact the Earth's radiative balance both directly by scattering light (Haywood et al. 1999; Lewis and Schwartz, 2004) and indirectly by affecting marine cloud formation (Clarke et al. 2006). At high wind speeds sea spray also has the potential to influence the air-sea transfer of heat and gases (e.g. Andreas et al. 1995; Jeong et al. 2012). From the perspective of atmospheric chemistry, sea spray droplets provide an important surface or medium





for heterogeneous reactions (Sievering et al. 1991; Kim et al. 2014). In coastal regions where roughly half of the world's

population resides, sea spray constitutes a major component of particulate matter in the marine atmospheric boundary layer.

Particulate matter is directly relevant for human health and is subject to air quality regulations.

Breaking waves entrain air into water, resulting in plumes of bubbles within the top meters of the ocean and the

appearance of whitecaps (Thorpe, 1992). Sea spray aerosols are primarily formed from the bursting of those bubbles. Two

modes of sea spray aerosols are commonly observed: the film drop mode (predominantly submicron) arising from disintegration

of relatively large bubbles, and the jet drop mode (≥500 nm to a few microns in radius) from relatively small bubbles (e.g.

Blanchard, 1963). Tearing of the wave crest under conditions of very high winds can also lead to large spume droplets (tens of

microns to millimeters; Monahan et al., 1983), which may contribute significantly to the total sea spray mass flux but not to the

number flux. The causal relationship between bubbles and sea spray has led to the application of the "whitecap" method for the

estimation of sea spray flux over the ocean: in which the spray production flux from unit area of a foam surface is prescribed

based on other measurements and scaled up by the whitecap fraction (e.g. Monahan et al. 1986; Martensson et al. 2003; Clarke et

al. 2006). However, the parameterization of whitecap fraction as a function of wind speed has an uncertainty of about an order

of magnitude in moderate winds (e.g. Anguelova and Webster, 2006).

Recent works suggest that the whitecap fraction is also sensitive to parameters such as sea state (Scanlon and Ward,

2016; Brumer et al. 2017a) and water temperature (Salisbury et al. 2014; Salter et al. 2015). In shallow waters, wave breaking

(and so whitecap fraction) depends not only on wind forcing, but also on the interactions between wind waves and swell with the

bottom topography (e.g. de Leeuw et al. 2000; Resio et al. 2002). The bubble number concentrations within the surf zone can be

about two orders of magnitude higher than over the open ocean (e.g. Brooks et al. 2009). van Eijk et al. (2011) estimated surf

zone aerosol flux on sandy beaches to be about an order of magnitude higher than for the open ocean under similar wind

conditions.

There are only a few datasets of direct measurements of sea spray fluxes by the eddy covariance (EC) method. Based

on data from an Arctic cruise, Nilsson et al. (2001) published the first EC measurements of aerosol number fluxes, which

correlated strongly with wind speed. Their data suggest that sea spray source flux consists of a film drop mode centered at ~50

nm radius and a jet drop mode center at 500 nm radius. Geever et al. (2005) measured submicron aerosol fluxes at radius > 5 nm

and > 50 nm at the coastal station of Mace Head on the west coast of Ireland. They found comparable aerosol number fluxes in

the Aitken mode (5–50 nm) and in the accumulation mode (50–500 nm). Norris et al. (2008) measured size-distributed aerosol

fluxes between 0.15 and 3.5 μm radius at the Duck Pier on the east coast of United States. They showed that sea spray flux

increases with the local wind speed up to a radius of 1 μm. More recently Norris et al. (2012, 2013b) measured size distributed

sea spray fluxes (0.18 < radius < 6.61 μm) over the open ocean, and explored the wind speed and wave Reynolds number



dependences of the flux. The wave Reynolds number (primarily a function of wind stress and significant wave height) was

found to explain about twice as much variance in measured open ocean sea spray fluxes than wind speed alone (Norris et al.

2013b).

Total and size distributed aerosol number fluxes have seldom been measured simultaneously, precluding an assessment

of the sea spray flux closure and distribution. Here we report concurrent measurements of "total" (radius > 1.5 nm) and size

distributed sea spray fluxes (about 0.1 < radius < 6 μm radius) from a coastal site over several months from February 2015 to

February 2017. We examine how sea spray fluxes vary with wind speed (1–21 m s$^{-1}$), significant wave height (0.2–3.1 m), wave

Reynolds number ($7\times10^3$ - $2\times10^6$) and other surface ocean parameters. We also examine the size distribution and closure in

aerosol number fluxes at different sea states.

## 2. Experimental

The Penlee Point Atmospheric Observatory (PPAO, http://www.westernchannelobservatory.org.uk/penlee/) on the

southwest coastal of the United Kingdom has proven to be a suitable site for eddy covariance measurements of air-sea transfer

(Yang et al. 2016a, 2016b, 2019). PPAO sits about 11 m above mean sea level and a few tens of m away from the water's edge.

The eddy covariance (EC) system, including the fast aerosol sensors and a sonic anemometer (R3, Gill), is mounted on a mast at

about 18 m above mean sea level. An ultra-fine condensation particle counter detecting aerosols > 1.5 nm radius (CPC-3025A,

TSI) and a compact lightweight aerosol spectrometer probe (CLASP, Hill et al. 2008) providing size spectra for radii between ca.

0.1–6 μm at ambient humidity were employed at PPAO. The CPC was used for flux measurements between 24 February and 3

June 2015. The CLASP unit deployed between 17 February and 1 March 2015 had a size range of 0.11 to 6 μm. This

encompasses a one-week of overlap with the CPC, which we primarily use to study the closure of aerosol number fluxes. A

slightly different CLASP unit was deployed between 21 December 2016 and 16 February 2017, which had a size range of 0.15 to

6 μm. Because of the narrower size range of this second unit and thus more undetected film drop aerosols, data from the 2016-

2017 period were less representative of the total number fluxes and only used to contrast between open water and fetch-limited

conditions.

Previous EC observations of momentum, sensible heat, $CO_2$, and $CH_4$ fluxes at PPAO show that two wind sectors are

representative of air-water transfer: the southwest sector for which airflow is from the open ocean, and the northeast sector with

airflow from the fetch-limited Plymouth Sound (Yang et al. 2016a; 2016b; 2019). A flux footprint model for spatially

homogeneous conditions (Kljun et al. 2004) predicts that under typical southwesterly winds, the majority of the turbulent flux at

a sensor height of 18 m above mean sea level comes from waters several hundred meters upwind of the site with a mean water





depth of ~20 m.  When winds are from the northeast, the flux footprint is over the Plymouth Sound and does not overlap with

land on the opposite side of the Sound (5–6 km away) except possibly under strongly stable conditions.

In the eddy covariance method, aerosol number concentration ($C$) measured at high frequency (here 10 Hz) is correlated

with the vertical wind velocity component and averaged over time to yield the net aerosol flux ($= \overline{C'w'}$, where primed quantities

are perturbations from the mean and the overbar is the averaging operator).  The measured net flux is the sum of the source

(upwards, positive) and deposition (downwards, negative) flux components (Nilsson et al. 2001; Geever et al. 2005; Norris et al.

2013b).  The sea spray source flux must be derived from the net flux by subtracting a modeled deposition flux ($= \overline{C \cdot V_d}$, where

$V_d$, is the aerosol deposition velocity).  Both the CLASP and the CPC fluxes are initially computed as 10-minute averages and

then filtered for non-stationary turbulence conditions following Yang et al. (2016a).  Valid fluxes can be averaged to hourly

intervals to reduce noise.

For the CLASP, a small fraction of the aerosols are lost to the short inlet (~25 cm) during the 0.1 s of transit time.  The

loss is size dependent and predicted inlet efficiency varies from essentially unity for aerosols smaller than 2 μm radius (film

droplets and most of the jet droplets), to ~0.5 at 5 μm, and to ~0.1 at 8 μm (Pui et al. 1987).  Spume aerosols are likely too large

to be efficiently sampled by the CLASP.  We corrected for aerosol loss in the CLASP inlet prior to the flux calculations.

Following Norris et al. (2012), the CLASP measurements are converted from net fluxes to source fluxes using the size-

distributed deposition velocities of Slinn and Slinn (1980).   Integrated over all CLASP sizes, this deposition correction amounts

to 25 cm$^{-2}$ s$^{-1}$ in the mean (up to ~200 cm$^{-2}$ s$^{-1}$) when winds were from the southwest (14% of the net flux on average and

occasionally over 50%).  For submicron aerosols only (i.e. radius of 0.1–0.5 μm), the deposition correction is 7 cm$^{-2}$ s$^{-1}$ in the

mean.  Humidity flux induces a bias in the CLASP aerosol flux (Fairall et al. 1984).  This is because the particles are sized at

ambient humidity, but grow/shrink with local relative humidity (RH).  We correct for this bias using the modified bulk correction

scheme described by Sproson et al. (2013).  The final CLASP fluxes are presented at a constant relative humidity of 80%

following the aerosol growth rate for sea salt reported by Gerber (1985).  Together these corrections amount to typically 20–30%

of the total CLASP source number flux at PPAO.  A robust humidity observation was unavailable from December 2016 to

February 2017.  The CLASP measurements during that phase could not be fully corrected for humidity effects and were thus

more uncertain.

The ultra-fine CPC sub-sampled from a Teflon tube (18 m long, 0.64 cm ID, flow rate of ~15 liters per minute).  The

use of Teflon tube is generally not recommended for aerosol measurements, as its non-conductive nature can lead to significant

aerosol losses; it was employed here out of convenience because the CPC was sub-sampling off of an existing inlet for $CO_2$ flux

measurements (Yang et al. 2016b).  A brief test of CPC measurements at PPAO between using the long Teflon inlet tube and a

~5 m, 0.32 cm ID stainless steel tube did not show any obvious difference in the aerosol number concentration.  The CPC


cospectra also do not suggest severe aerosol losses at the high frequency end. The better than expected transmission through the Teflon sampling tube may be because the tube had been used under high flow rate at PPAO for nearly a year prior to the CPC

measurements. It was well coated with sea salt, which probably increased its conductivity and thus reduced electrostatic aerosol losses.

The long sampling tube resulted in a delay time of ~3 s in the CPC signal relative to the turbulent wind measurements, as determined by the maximum covariance method between aerosol concentration and vertical wind velocity. This delay is close to the expected time based on inlet length, diameter, and flow rate in the main tube (2.3 s). The CPC inlet efficiency, accounting

for its length, a 90° turn and three bends (not including any electrostatic losses), is predicted to be essentially unity for 0.3 < radius < 2 μm (Pui et al. 1987). For radius < 0.3 μm, the mean inlet efficiency is 0.96 (<0.8 for nucleation mode aerosols). For radius > 3 μm, the efficiency drops to ~0.7. Previous observations in the marine atmosphere show that total aerosol number is usually dominated by particles below a radius of about 0.3 μm (e.g. Hoppel et al. 1990), which suggests only a minor inlet loss for the CPC ($\leq$ 4%). We choose not to apply any inlet efficiency correction to the CPC flux since the corrections are probably

small and the size distribution of aerosol flux below 0.1 μm (lower cutoff of CLASP) is not known. We correct our CPC fluxes for high frequency flux attenuation due to the finite instrument response time (1 s for 95% change) using an empirical filter function approach analogous to that for gas flux measurements (e.g. Yang et al. 2013). Increasing with wind speed, this correction amounts to ~10% of the CPC number flux for the average condition at PPAO.

The deposition correction for the CPC flux is only approximate due to both uncertainties in $V_d$ and the lack of

knowledge of the fine/Aitken mode aerosol size distribution. The deposition velocity for submicron aerosols over water, dependent on aerosol size and environmental conditions, is on the order of approximately 0.01–0.1 cm s$^{-1}$ (Slinn and Slinn 1980; Duce et al. 1991). The aerosol size distribution below 100 nm radius was not measured at PPAO but previous maritime observations of submicron aerosols generally suggest peak number concentrations at approximately R= 25 and R= 100 nm (e.g. Hoppel et al. 1990). The Slinn and Slinn (1980) parameterization predicts $V_d$ of 0.034 and 0.010 cm s$^{-1}$ for these two aerosol

sizes for the mean conditions at PPAO. For simplicity, we take the average of the two $V_d$ values (0.022 cm s$^{-1}$ at the mean wind speed at PPAO) and multiply it by the CPC number concentration to estimate the deposition flux. When winds were from the southwest, the deposition flux amounts to 33 cm$^{-2}$ s$^{-1}$ in the mean (19% of the net flux).

Frequency weighted cospectra of total CLASP and CPC net aerosol fluxes averaged to wind speed bins are shown in Figures 1 and 2 (for the open water wind sector). In low-to-moderate winds, the aerosol cospectra are broadly consistent with

the theoretical spectral shape for turbulent transfer (Kaimal et al. 1972) and with previously observed gas cospectra at PPAO (Yang et al. 2016a). With increasing wind speeds the magnitudes of the cospectra increase, reflecting greater sea spray fluxes. The CPC cospectra are much noisier than those of the CLASP. This may be because most of the aerosols detected by the


CLASP arise from sea spray emission. In contrast, only a very small fraction of the aerosols detected by the CPC participate in rapid air-sea exchange (i.e. sea spray emission); the vast majority comes from other sources (e.g. pollution) and increases the

random measurement noise in the CPC flux (see analogous discussion on methane flux by Yang et al. 2016a).

## 3. Results and Discussions

### 3.1 Sea Spray Flux Closure and Wind Speed Dependence

Aerosol fluxes were almost always positive (i.e. from sea to air), indicating sea spray emission. Figure 3 shows the time

series of total aerosol number concentrations and source fluxes from the CPC and the CLASP (during the one week of overlap), significant wave height and tide height above datum, as well as wind speed and wave Reynolds number. Winds were coming from the southwest for the majority of this week with peak speed $> 16$ m s$^{-1}$. The CPC detected baseline concentrations of a few hundred aerosols cm$^{-3}$, typical for a marine atmosphere. Many short and sharp spikes (of the order of thousands cm$^{-3}$) are apparent in the CPC time series. These spikes are often coincident with spikes in sulfur dioxide and carbon dioxide from ship

exhaust emissions. CPC fluxes during such brief periods of excessive variability (i.e. relative standard deviation over 50%) are removed on the basis of non-stationarity.

Aerosol number concentration summed over all sizes is much lower and more constant from the CLASP. For the southwest wind sector, the median supermicron and submicron aerosol concentrations detected by the CLASP were 11 and 24 cm$^{-3}$ respectively. The magnitude of the former is consistent with the typical sea spray aerosol number concentration in the

marine boundary layer (e.g. O'Dowd and Smith 1993). The CLASP did not detect ship plumes as ship emitted particles tend to be small ($< 30$ nm radius) and below the measurement size cutoff (e.g. Hobbs et al., 2000; Petzold et al., 2008). Despite the large differences in total number concentrations between the CPC and CLASP, aerosol fluxes from the two instruments were generally comparable in magnitude. The CPC fluxes were slightly higher than the total CLASP number fluxes by a mean difference of 45 cm$^{-2}$ s$^{-1}$ during this week. As discussed in Section 3.3, this difference is likely due to the part of the film drop

mode not completely captured by the CLASP as a result of its lower size cutoff ($\sim 0.1$ μm).

Sea spray flux peaked during periods of high winds and large waves. At low tide, the distance between the water's edge and the flux sensors increases, shifting the center of the flux footprint closer to the shoreline. Previously observed transfer rates of momentum, sensible heat, and gases ($CO_2$ and $CH_4$) did not vary with the tidal height (Yang et al. 2016a, 2016b), suggesting that the narrow surf zone (width of a few m) beyond the rocky shoreline in front of PPAO has a negligible influence on the

measured fluxes. The same appears to be true for the aerosols, as periods of low tide do not consistently result in large sea spray fluxes (Figure 3, Panels B&C). Thus unlike Geever et al. (2005), we do not need to filter out data during low tide conditions.





On 27 February 2015 during light, northerly winds from over the land, aerosol fluxes from both instruments were near zero, as expected.

Figure 4 shows the wind speed dependence in aerosol source flux from the CPC (February to June 2015) and from the
CLASP (February to March 2015). Here we have restricted data to the southwest wind sector (open water) only. Across most of the wind speed range, total aerosol source fluxes from the CPC and the CLASP show a similar relationship with wind speed. Total sea spray source flux from PPAO amounts to about 200 $cm^{-2}$ $s^{-1}$ in the mean (median of ~150 $cm^{-2}$ $s^{-1}$), increasing non-linearly with wind speed up to ~1000 $cm^{-2}$ $s^{-1}$ at a wind speed of 20 m $s^{-1}$. These fluxes are in reasonable agreement with the source flux relationship found by Geever et al. (2005; >5 nm radius) for a coastal site at Mace Head during high tide, when wave
breaking at the shore did not unduly influence their measurements. In comparison, the relationship found by Nilsson et al. (2001; >5 nm radius) from the Arctic Ocean is significantly higher than the PPAO measurements at wind speeds above ~12 m $s^{-1}$. Bin-averages of total aerosol number flux measurements at PPAO are plotted on a log scale against wind speed in Figure 5 along with the parameterizations from Nilsson et al. (2001) and Geever et al. (2005) for both net and source fluxes. Measured net fluxes at PPAO agree very well with the net fluxes from Geever et al. (2005). The source fluxes show greater discrepancy,
likely due to the different deposition correction schemes applied.

### 3.2 Non-Wind Speed Controlling Factors on Sea Spray Fluxes

Sea spray flux is more strongly associated with significant wave height ($Hs$, derived from the full wave spectrum) than with wind speed (Figure 6). Wave data when winds were from the southwest are taken from a Waverider buoy from Looe Bay,
about 16 km west of PPAO (http://www.channelcoast.org/data_management/real_time_data/charts/?chart=98). The stronger dependence on $Hs$ is reflected in the hourly CPC data (n = 452) by both higher $R^2$ (0.65) as well as the Spearman's rank correlation coefficient (R = 0.71) compared to the dependence on wind speed ($R^2$ = 0.47 and Spearman's R = 0.51). When $Hs$ was below 0.5 m, the mean CPC source flux averaged to about zero. Such a strong dependence of sea spray flux on wave height is typically not observed over the open ocean (e.g. Norris et al. 2013b), and is likely due to coastal wave breaking. Equilibrium
wind waves at wind speeds > 10 m $s^{-1}$ as well as swell have wavelengths that are longer than twice the water depth within the PPAO flux footprint (Pierson and Moskowitz, 1964), and should shoal due to interactions with the sea floor. We thus expect coastal waves to break more frequently and generate more sea spray compared to open ocean waves at the same wind speeds.

Following Zhao and Toba (2001), we computed the wave Reynolds number as $R_{Hw} = u_* H_s / \nu$, where $u_*$ is the friction velocity from eddy covariance and $\nu$ is the kinematic viscosity of seawater. Sea spray flux increases approximately linearly with
the wave Reynolds number ($R_{Hw}$, Figure 7). The source flux dependence on $R_{Hw}$ is slightly weaker than on $Hs$ but stronger than on wind speed ($R^2$ = 0.62 and Spearman's R = 0.63) for the open water sector. Norris et al. (2013b) found a linear dependence





between sea spray flux and $R_{Hw}$. They also argued that below a critical $R_{Hw}$ of 7.2e$^4$, wave breaking does not occur and sea spray source flux should be zero, in general agreement with observations here. The linear dependence of sea spray flux at PPAO on $R_{Hw}$ is qualitatively consistent with their open ocean results. The $R_{Hw}$ parameterization from Norris et al. (2013b) integrated over

all CLASP size bins is shown in Figure 7. We see that sea spray fluxes measured at PPAO exceed those open ocean observations by about an order of magnitude (but are smaller than estimates over a surf zone by Clarke et al., 2006, as shown in Section 3.3).

The influence of waves on coastal sea spray generation is further illustrated in the comparison between open water and fetch-limited conditions (Figure 8). We use data from the second CLASP deployment (December 2016 to February 2017) for

this analysis, as winds were seldom from the northeast during the first CLASP deployment (February-March 2015). Here we separate the total sea spray fluxes measured by the CLASP into two different wind sectors: the southwest (open water) and the northeast (facing the Plymouth Sound with a fetch over water of ~5 km). At a given wind speed, sea spray fluxes were generally greater for the open water sector than for the fetch-limited sector. Wind speed was an even poorer predictor of sea spray flux from the open water during this period, as some high sea spray fluxes were observed at low wind speeds due to the presence of

large swell. Fluxes from both wind sectors show better correlations against $H_s$, though with different trends; higher fluxes are observed for the fetch-limited conditions than for open water at a given wave height. Here $H_s$ in the Plymouth Sound is predicted using a parameterization for fetch-limited waters (Resio et al. 2002) since a direct measurement is not available. The different functional dependencies of the sea spray flux on wind speed and wave state for open water and fetch-limited conditions are reconciled when their joint influence is accounted for by the wave Reynolds number. When plotting against $R_{Hw}$ sea spray

fluxes from both wind sectors fall closely on the same curve (Figure 8C). The importance of jointly accounting for the influence of both wind and waves during air-sea exchange processes is well illustrated by the strong correlation between $R_{Hw}$ and sea spray flux. This result is consistent with the recent findings of Brumer et al. (2017a,b) for whitecaps and gas transfer, and from Norris et al. (2013b) for sea spray fluxes.

There is an observable sea-state dependence in the sea spray flux for the open water sector. Here we separate the CPC

flux data into two groups of sea states: increasing wind speed (i.e. hourly increase by more than 1 m s$^{-1}$) and decreasing wind speed (i.e. hourly decrease by more than 1 m s$^{-1}$). These correspond approximately to younger/growing seas, and older/more developed seas, respectively. As shown in Figure 9, at intermediate wind speeds (~10 m s$^{-1}$) the aerosol flux is in the mean about twice as high during periods of decreasing wind speed than during periods of increasing wind speed, qualitatively consistent with Norris et al. (2013b) for sea spray fluxes and Callaghan et al. (2008) for whitecap fraction. $H_s$ is also larger during periods of

decreasing wind speed. Waves shoal and are more likely to break near the coast regardless of their state of development. Thus larger waves from more developed seas tend to lead to greater sea spray fluxes in these kinds of coastal environments.


It is interesting that there is a large discrepancy in magnitude in the source flux vs. $R_{Hw}$ relationship between the open ocean observations from Norris et al. (2013b) and those at PPAO (Figure 7) — if $R_{Hw}$ is such a good predictor of air-sea fluxes, why does it fail to reconcile these two data sets? We suggest two possible reasons. First, shoaling may result in more frequent

and intense wave breaking near the coast compared to the open ocean as the waves steepen upon approaching the shore (e.g. Elgar et al., 1997). Second is a potential difference in aerosol production per unit area whitecap between a coastal region with shoaling waves and the open ocean as a result of the different wave breaking and bubble generation processes (Deane and Stokes, 1999; Lewis and Schwartz, 2004; de Leeuw et al. 2011). Bubble populations near the sea surface were generally found to be higher close to the coast than in open water (Johnson and Cooke, 1979; Brooks et al. 2009). Different void fractions have

also been observed beneath plunging and spilling breaking waves (Rojas and Loewen, 2010). Testing of these hypotheses requires observations of the relationships between whitecaps, bubbles and $R_{Hw}$ in the near-shore region.

Laboratory measurements suggest dependences in sea spray flux on water temperature and salinity (e.g. Martensson et al. 2003; Salter et al. 2014), while Tyree et al. (2007) showed that the addition of natural organic matter increased the submicron aerosol flux by 50%. Previous observations of aerosol composition in marine environments imply that a significant fraction of

sea spray is made up of organic materials (e.g. O'Dowd et al. 2004). We investigate these dependencies by comparing our sea spray flux data to surface ocean parameters from the marine station L4 (6 km south of PPAO) of the Western Channel Observatory (http://westernchannelobservatory.org.uk/). From February to June 2015, sea surface temperature and salinity varied between 9.2 and 12.5 °C and between 35.1 and 35.3, respectively. Chlorophyll a increased from 0.8 to 3.1 mg m$^{-3}$ during the spring phytoplankton bloom, while colored dissolved organic matter (from the E1 buoy 18 km offshore) also varied by about

a factor of four. We test the importance of these surface ocean parameters by examining their correlations vs. $F_{SSA}'$, where $F_{SSA}'$ is the total sea spray source flux ($F_{SSA}$) minus a polynomial fit to $F_{SSA}$ (as a function of wind speed or $H_s$). No significant correlation was found. This suggests that within the range of conditions observed at PPAO during this deployment, waves and wind are the first-order drivers for sea spray formation, while the other parameters appear to be of little importance.

**3.3 Size Distributed Aerosol Number Concentrations and Fluxes**

Figure 10 shows number distribution (dN/dR$_{80}$) vs. radius at a humidity of 80% (R$_{80}$) averaged in wind speed as well as significant wave height bins. Data are taken from February–March 2015 and for the southwest (open water) wind sector only. The overall distribution in concentration is fairly typical of the marine atmosphere, with most aerosols in the submicron mode. The number distributions are more clearly segregated by significant wave height than by wind speed for radius up to about 2 μm.

Above this cutoff, the number distribution seems to be largely independent of $H_s$ or wind speed.


In accordance with the total aerosol number fluxes, size distributed fluxes from the PPAO ($dF/dR_{80}$) are significantly higher than measurements over the open ocean (Figure 11). For example, in moderate seas at PPAO the measured source flux at $R_{80}$ of 1 μm exceeds $10^6$ m$^{-2}$ s$^{-1}$ μm$^{-1}$, compared to approximately $10^5$ m$^{-2}$ s$^{-1}$ μm$^{-1}$ from Norris et al. (2013b). Sea spray flux from the coastal seas near PPAO is lower than estimates from the surf zone by Clarke et al. (2006). Readers interested in other

previous measurements and parameterizations of size distributed sea spray source fluxes are referred to reviews by O'Dowd and de Leeuw (2007) and de Leeuw et al. (2011).

Size distributed aerosol number concentrations as well as fluxes peak at the lowest size bin of CLASP (0.1 μm radius at ~80% RH), near the typical mode center for film droplets (Martensson et al. 2003; Tyree et al. 2007). We can crudely estimate the contribution of film drop fluxes below the detection cutoff for the CLASP by fitting a log-normal distribution to the observed

$dF/dR_{80}$. Here we assume that the observed $dF/dR_{80}$ at a $R_{80}$ of 0.1 μm represents the peak in the film drop mode. Integrating the log-normal fit from 10 nm to 0.1 μm yields the "missing" film drop flux (median value of ~40 cm$^{-2}$ s$^{-1}$), which amounts to about 25% of the total measured CLASP number flux. This is consistent with the finding that the total aerosol number flux from the CLASP was lower than that from the CPC by an average of 45 cm$^{-2}$ s$^{-1}$.

About 70% of the sea spray fluxes measured by CLASP are submicron (at RH of 80%), with the vast majority of the

aerosol number flux residing between radius 0.1 and 1.1 μm (Figure 11). Aerosols with radius greater than 2 μm make up only ~1% of the total CLASP number flux. The distribution of source number flux below a radius of about 2 μm is more clearly segregated by $H_s$ than by wind speed, as with the aerosol number size distribution. Above 2 μm radius, size distributed fluxes no longer seem to depend on $H_s$ or wind speed. There is a subtle decreasing trend in the ratio between film drop mode (radius of ~100 nm at 80% RH) and jet drop mode (radius of ~500 nm at 80% RH) with increasing significant wave height (see Figure 12).

An examination of normalized $dF/dR_{80}$ (by the respective mean flux) shows that with greater wave height, the jet drop mode appears to increase more steeply than the film drop mode. A similar shift in the shape of size distributed aerosol flux with growing seas has been observed previously by Norris et al. (2013a). They showed that the bubble spectra change with wind speed, with the concentrations of small bubbles (jet mode aerosols) increasing more rapidly than the large bubbles (film drop mode aerosols).


## 4. Concluding Remarks

Eddy covariance measurements of sea spray fluxes from the coastal site Penlee Point Atmospheric Observatory (PPAO) show that about 70% of the total detected number fluxes were submicron. A reasonable closure is found between the aerosol number flux from the CPC (>1.5 nm in radius) and the total number flux from the CLASP (0.1–6 μm in radius) after considering

the incomplete detection of film mode aerosols by the latter. Sea spray fluxes from the open water wind sector at PPAO increase



with wind speed with a dependence that is similar to previous coastal sea spray flux measurements at Mace Head (Geever et al. 2005). Our observed fluxes are greater in magnitude than most previous open ocean measurements except those reported by Nilsson et al. (2001).

Sea spray formation in this coastal environment is strongly dependent on sea state. Both significant wave height ($H_s$)

and wave Reynolds number ($R_{Hw}$) are better predictors of sea spray fluxes than local wind speed. The importance of waves is further confirmed by comparing sea spray fluxes measured from the open water sector to fluxes from the fetch-limited Plymouth Sound, where waves were much smaller at a given wind speed. For both wind sectors, sea spray fluxes correlate with $H_s$ more strongly than with wind speed, but with two very different relationships. The wave Reynolds number ($R_{Hw}$) reconciles the fluxes from the two sectors. This finding is consistent with those of Brumer et al. (2017a,b), who found $R_{Hw}$ to be a much better

predictor than wind speed of both the whitecap fraction and gas transfer velocity in different wind/wave regimes.

Sea spray fluxes measured at PPAO (open water sector) are likely only representative of the nearest few km(s) from shore. The median supermicron and submicron sea spray fluxes from the CLASP during February–March 2015 were about 60 and 100 cm$^{-2}$ s$^{-1}$, respectively. The residence times of supermicron and submicron aerosols in a 500 m deep marine atmospheric boundary layer (MABL) against dry deposition are of the order of 0.6 and 6 days (at a respective deposition velocity of 1 and 0.1

cm s$^{-1}$, Slinn and Slinn, 1980). The residence time for submicron aerosols is likely further reduced by wet deposition (~3 days; Lewis and Schwartz, 2004). At these timescales, the expected steady state supermicron and submicron aerosol number concentrations for a well-mixed MABL based on our measured fluxes would be on the order of 60 cm$^{-3}$ and 300 cm$^{-3}$, respectively. These are significantly higher than the observed median supermicron and submicron aerosol concentrations from the CLASP of 11 and 24 cm$^{-3}$ for the open water wind sector. Clearly aerosol concentrations and fluxes are not in steady state in

this coastal environment. The fluxes are enhanced near the coast due to increased wave breaking resulting from shoaling of waves in shallow water, while the concentrations reflect both sea spray generated within the flux footprint as well as the aerosol sources and sinks further upwind. To map out the spatial distributions of sea spray fluxes, measurement techniques such as eddy covariance from a ship and aerial imaging of whitecap fraction (e.g. from an unmanned aerial vehicle) are needed.

**Data availability**

The data collection for this work was unfunded and thus the data will not be submitted to a central depository. Please contact the corresponding author for the data.



**Author contribution**

MY carried out the majority of the measurements. IB provided the CLASP and CPC instruments. IB and SN helped with

instrument installation and data analysis. TB helped with maintenance of the instruments at PPAO. MY prepared the

manuscript with contributions from all co-authors.

**Competing Interest**

The authors declare that they have no conflict of interest.


**Acknowledgements**

Trinity House owns the Penlee site and has kindly agreed to rent the building to PML so that instrumentation can be protected

from the elements. We are able to access the site thanks to the cooperation of Mount Edgcumbe Estate. We thank J. Prytherch

and M. Salter (Stockholm University) for useful discussions, F. Hopkins, T. Smyth and P. Nightingale (Plymouth Marine

Laboratory, PML) for support, R. Pascal and M. Yelland (National Oceanography Centre, Southampton) for the loan of the R3

sonic anemometer, and B. Carlton (PML) for setting up data communication. The wave data from Looe Bay are provided by the

Channel Coastal Observatory. The Penlee site is part of the Western Channel Observatory, which is funded by NERC's National

Capability programme. This work is a contribution to the ACSIS project (The North Atlantic Climate System Integrated Study;

NE/N018044/1), and is contribution number 6 from the Penlee Point Atmospheric Observatory.

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




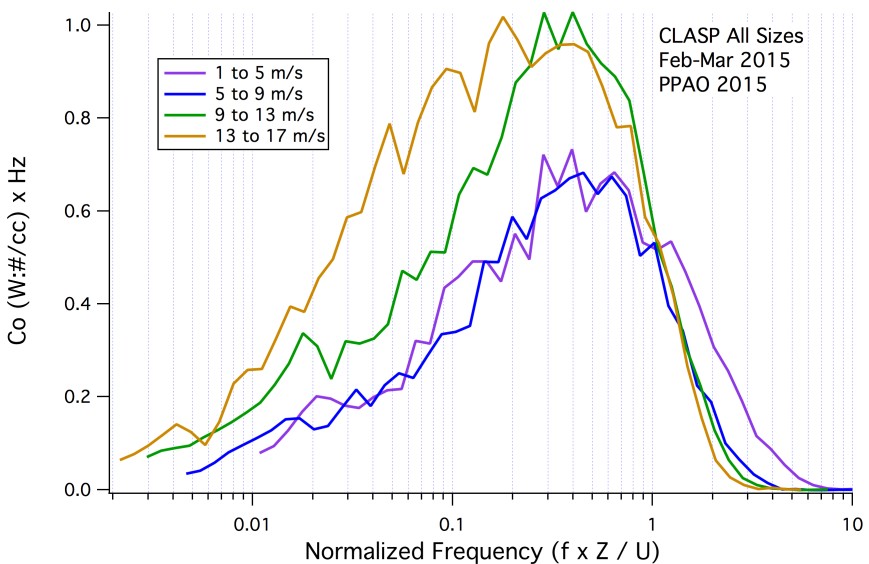

Figure 1. Averaged CLASP frequency-weighted total (net) aerosol number cospectra in wind speed bins for the southwest (open water) wind direction.

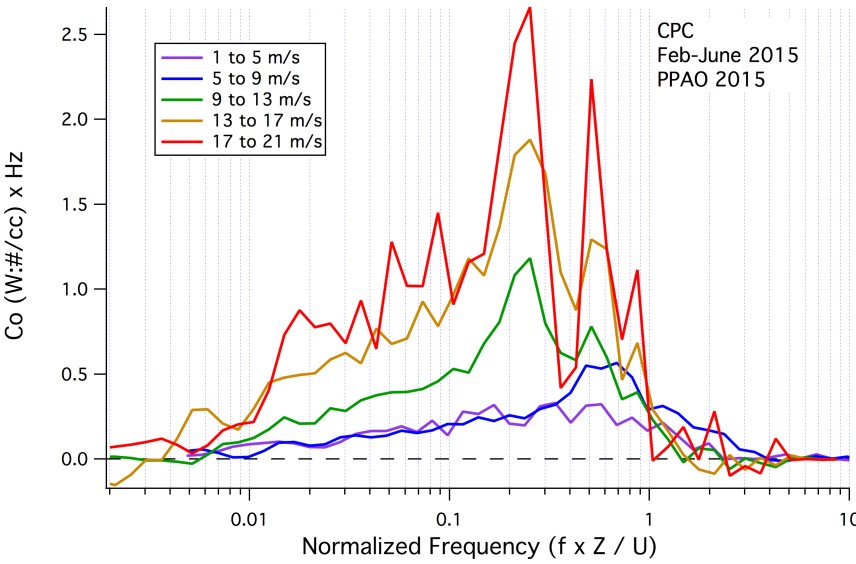

Figure 2. Averaged CPC frequency-weighted (net) aerosol number cospectra in wind speed bins for the southwest (open water) wind direction.







Figure 3. Times series of (A) CPC and CLASP total aerosol number concentration, (B) total aerosol number source flux from the CPC and CLASP, (C) significant wave height and tide height, and (D) wind speed (color-coded by wind direction) and wave Reynolds Number.



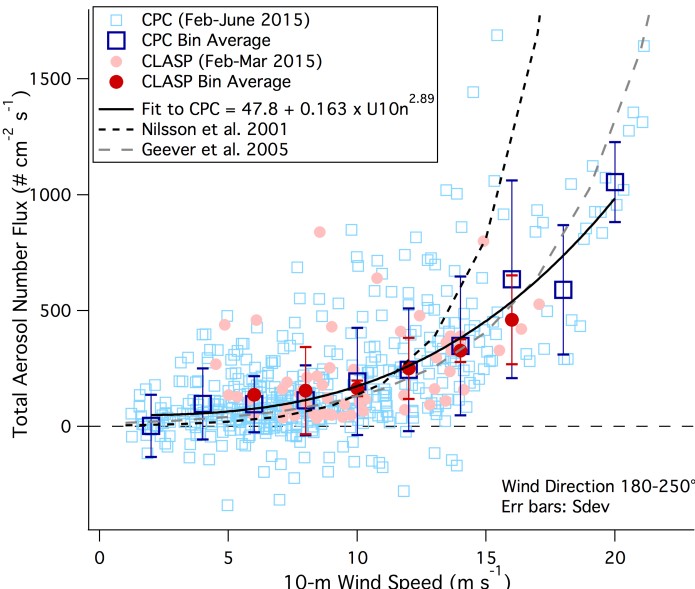

Figure 4. Hourly aerosol number source fluxes from the CPC and CLASP vs. wind speed (data from 2015). Error bars correspond to standard deviations within the bins. Also shown are source flux relationships derived by Geever et al. (2005; > 5 nm radius) from the coastal site of Mace Head and Nilsson et al. (2001; > 5 radius nm) from the open ocean of the Arctic.

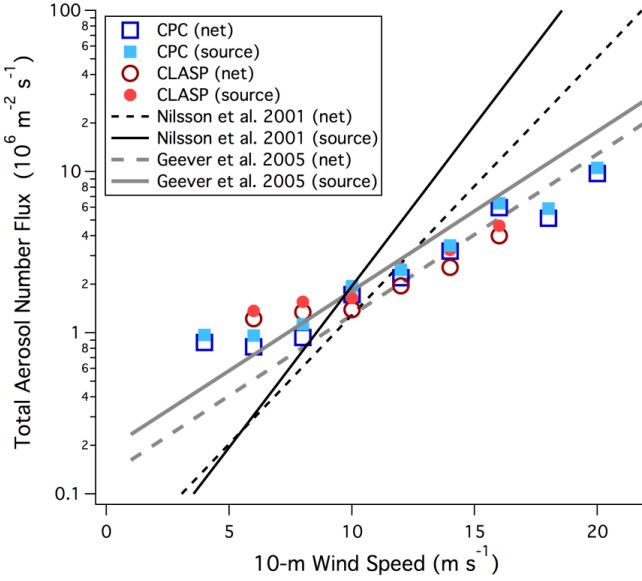

Figure 5. Bin-averaged aerosol number fluxes (net and source) from the CPC and the CLASP (in log scale) vs. 10-m wind speed (data from 2015). Also shown are net and source flux relationships derived by Geever et al. (2005; > 5 nm radius) from the coastal site of Mace Head and Nilsson et al. (2001; > 5 radius nm) from the open ocean of the Arctic. Error bars (2 x standard error) are smaller than the marker size and thus not displayed.





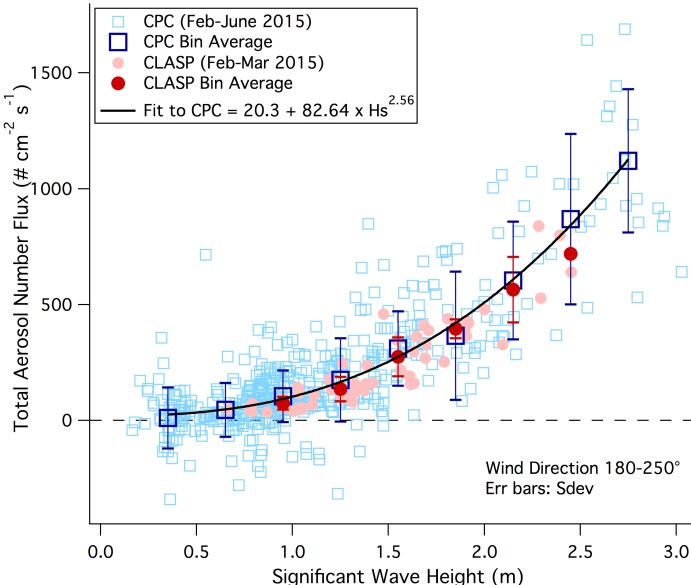


Figure 6. Hourly aerosol number source fluxes from the CPC and CLASP vs. significant wave height (data from 2015).

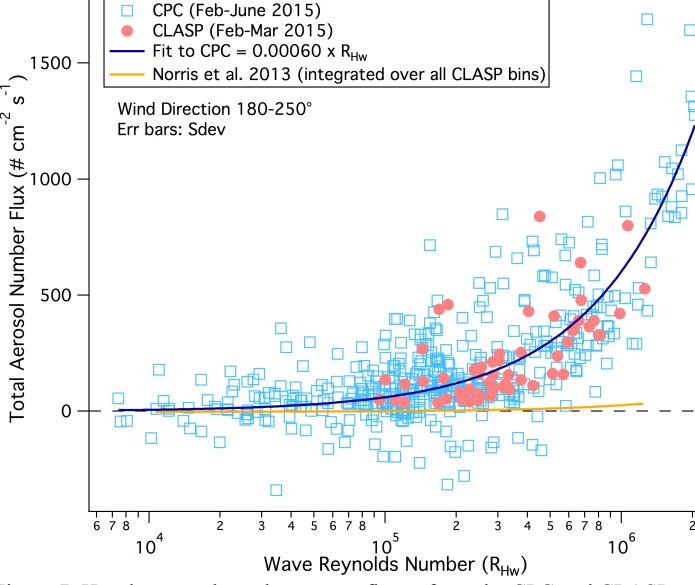


Figure 7. Hourly aerosol number source fluxes from the CPC and CLASP vs. Wave Reynolds Number (log scale; data from 2015). Also shown are a linear fit to the CPC fluxes and the Reynolds Number parameterization from Norris et al. 2013 (integrated over all CLASP size bins).




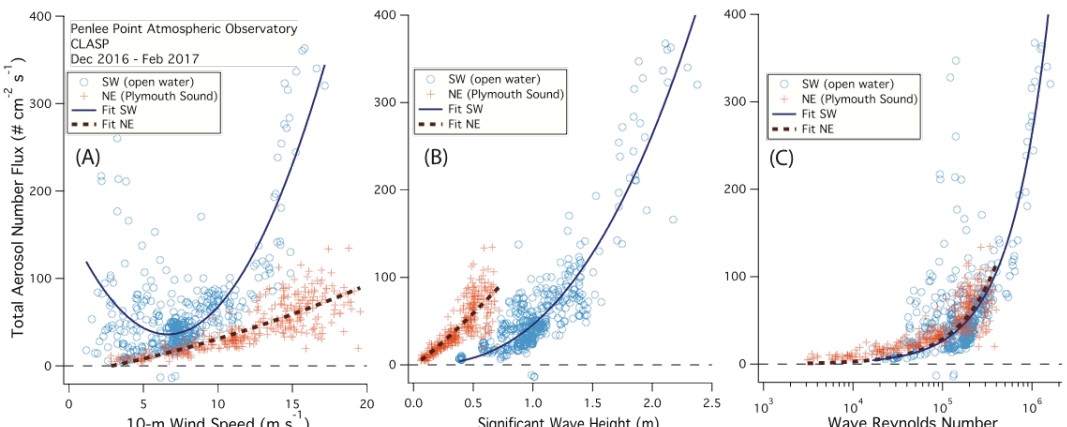

Figure 8. Total aerosol number flux from the CLASP (December 2016–February 2017) vs. wind speed (A), significant wave height (B), and wave Reynolds number (C). Data are separated into two distinct wind sectors: the southwest sector that faces the open water, and the northeast sector that faces the Plymouth Sound (fetch of ~5 km). Power fits of aerosol fluxes to $H_s$ and linear fits to $R_{Hw}$ are also shown.






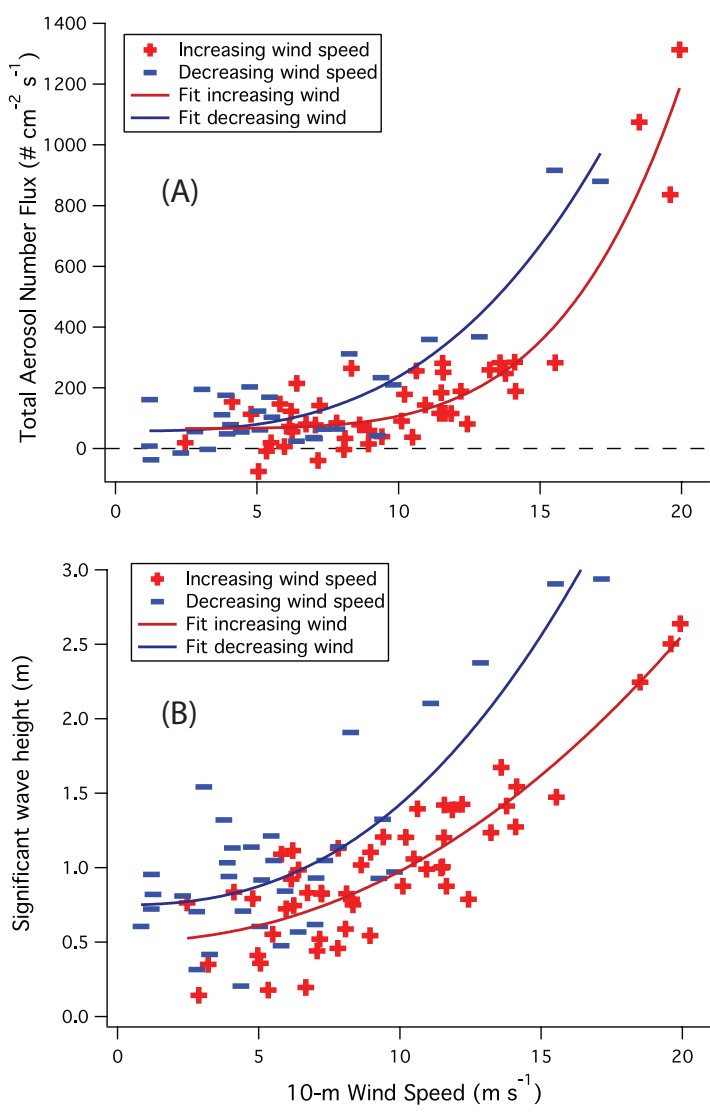

Figure 9. Aerosol number flux from the CPC for the southwest, open water wind sector versus wind speed (A) and versus
significant wave height (B). Data on both plots are separated into periods of increasing wind speed (red crosses) and decreasing
wind speed (blue dashes, see Section 3.2 for detail).





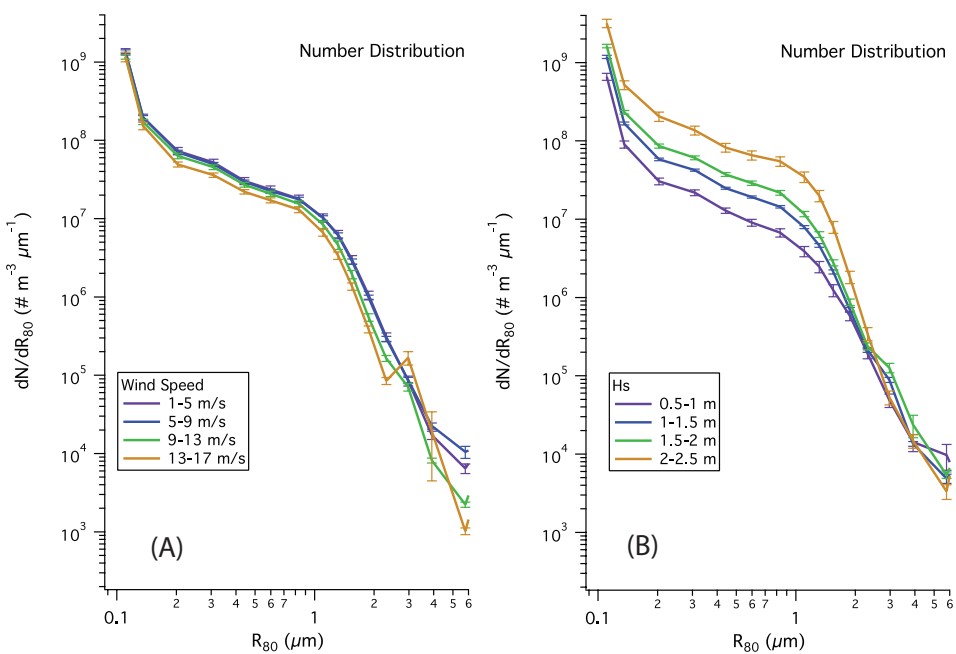

Figure 10. Size distributed aerosol number from the CLASP, bin-averaged according to wind speed (A), and significant wave height (B). Data are limited to the southwest (open water) wind sector only. Error bars indicate standard errors. Radius adjusted from ambient humidity to a relative humidity of 80%.

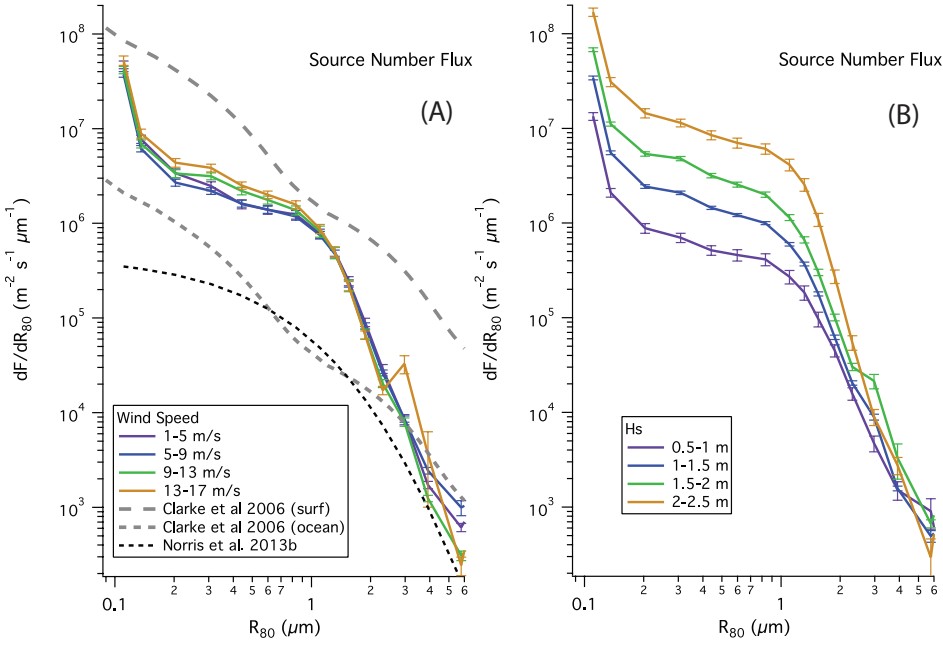

Figure 11. Size distributed source number flux from the CLASP, bin-averaged according to wind speed (A), and significant wave height (B). Data are limited to the southwest (open water) wind sector only. The open ocean source flux parameterizations from Clarke et al. (2006) and Norris et al. (2013b) are approximated at a wind speed of 10 m s$^{-1}$, while the measured surf zone flux from Clarke et al. (2006) is more than an order of magnitude higher. Error bars indicate standard errors. Radius adjusted from ambient humidity to a relative humidity of 80%.





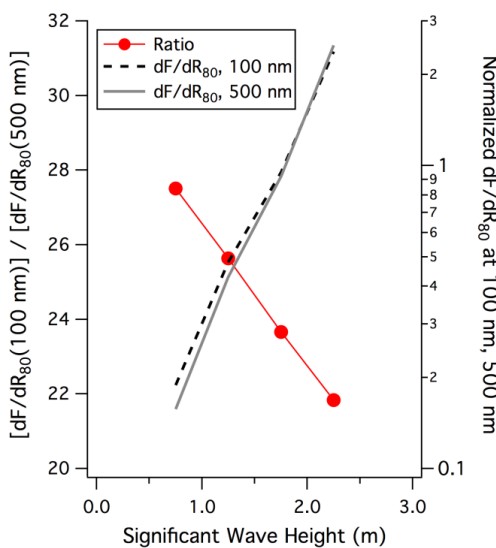

Figure 12. Ratio of size distributed source number flux in film drop mode ($R_{80}$ = 100 nm) to that in the jet drop mode ($R_{80}$ = 500 nm) as well as normalized dF/dR$_{80}$ at these sizes as a function of significant wave height.