# Peer review of "Sea Spray Fluxes from the Southwest Coast of the United Kingdom –"

_Atmospheric Chemistry and Physics, 2019_

## Referee Comment (RC1) · Christopher Fairall (Referee) · 8 Oct 2019

This paper is a description of an analysis of aerosol measurements made at Penlee Pt (PPAO ) on the coast of the UK. Eddy correlation aerosol fluxes are computed and are interpreted as the sum of source and deposition fluxes. An effective aerosol source strength is computed and analyzed with considerations of wind direction, etc.. The authors find the source strength correlations better with wave height and/or wave parameters than with wind speed. The source is stronger than expected for open ocean but weaker than that observed directly from a surf zone. The interpretation is that wave breaking is more intense (or something) in a shallow zone close to shore than the open

ocean. Most of the number flux occurs for aerosols greater than .1 and less than 1 micron. The paper is well written and the authors have carefully considered a number of experimental and physical aspects of the analysis and interpretation. In my view it can be published in its present form. I have a few comments the authors may with to consider.

*I suggest the authors make it painfully clear that their results are not affected by surf generation and the enhanced production they see is associated with enhanced breaking in shallow water but external to the shore break. Maybe they thought it was obvious but I pondered this.

*I suggest they carefully check terminology of net, source, and total aerosols. I found myself wondering if they were consistent. A number of figures say 'total aerosol number flux' but I am not confident I know if it is net or source.

Line 91. I don't think the turbulent flux is the same as the net flux. To me, net=turb-VgC, where Vg is the gravitational fall velocity.

Net=Source-VdC=turb-VgC

For the sizes they are considering, it may be that Vg is much less than Vd. They should state this. If Vg is negligible, then Source=turb+VdC

On line 94 they state that source is obtained by subtracting deposition from net Source=turb-VdC

Doesn't seem consistent with Fig 5, where source is greater than net. Please check this and get it straight.

Also suggest they read Andreas et al. JGR, vol 115, C12065, 2010; Freire et al BLM 160:249, 2016; and Nissanka et al. JGR, 9688:9702, 2018.

Figure 11 is certainly interesting. It is surprising that aerosol spectral concentration and source flux is independent of wind speed. The graphs might be a little easier to

use of the vertical axis was multiplied by R (are conserving).

---

## Referee Comment (RC2) · Sebastian Landwehr (Referee) · 22 Oct 2019

This contribution describes total and size resolved Eddy Covariance measurements of aerosol number flux at a coastal research station. The total number flux measured with a CPC integrates particles larger than 1.5 nm. While size resolved fluxes between 0.1 and 6 um are derived from a CLASP. The authors combine the measured net fluxes with size resolved dry deposition velocities derived from the model of Slinn and Slinn (1980), in order to obtain the source flux over the footprint area. The dependence of the size resolved and size-integrated source flux on wind speed, wave height, and wave Reynolds number is studied in detail and compared with previous parametrizations

that where derived over the open ocean and in coastal waters. The authors conclude that their measurements and findings are representative for coastal waters (excluding the surf zone) where sea spray emissions are largely enhanced due to costal wave breaking. Eddy Covariance measurements of aerosol fluxes in the marine environment provide an almost direct way to measure sea spray emission fluxes, however due to the complexity of making these measurements only few data sets exist today. Quantification of coastal sea spray emissions are relevant to the aerosol concentration and composition in the densely populated coastal regions and may thus bear relevance for heath related studies. Comparing the relation of coastal and open ocean sea spray emissions to forcing parameters may help to further improve the understanding of sea spray emission in general. I therefore consider this paper to be a valuable contribution and recommend publication with minor revisions as outlined below.

*As stated by referee Christopher Fairall the source flux can be derived from the measured turbulent flux (FEC) via Source = FEC+Vd*C-Vg*C, where C is the particle concentration, and Vd and Vg are the dry deposition and the settling velocity respectively. Vg depends on the particle size, shape and density, and hence on the relative humidity and the hygroscopy of the particles. The deposition velocity Vd may in additional be affected by turbulent and diffusive processes. Various dry deposition parametrizations can be found in the literature. The authors settle for the model of Slinn and Slinn (1980), which assumes that the particles assume their dry diameter in the constant flux layer and grow to a diameter defined by the growth rate and the assumed equilibrium relative humidity above seawater. Depending on the ambient relative humidity, the assumption of dry sea salt particles in the constant flux layer may introduce some errors. I therefore recommend the authors to explicitly state how they apply Slinn and Slinn (1980), i.e. which diameter conversions are performed, what particle densities are assumed.

*I further suggest that the ambient relative humidity time series should be presented in Fig 3., since it should play an important role in determining the correction term (Vd*C-Vg*C).

Line 103 to 105: Please consider splitting this sentence into two. Maybe the time series of the correction factors could be provided in the appendix to illustrate the variability of the relative importance of the correction term.

Line 109: Please specify which of the formulations, provided in Gerber (1985), you are using and with which set of coefficients?

Line 115: please correct "off of"

Line 138: R (=radius) has not been defined previously. Theres is some chance of confusion with the coefficient of determination. At this note, please unify the use of italic and noraml font for variables like Hs and RHw.

Line 134 to 142: How much does Vd predicted by Slinn and Slinn (1980) vary over the range of conditions encountered during the observations? Would accounting for these variations reduce the scatter in Figures 4, 6, and 7? However, these variations may be small compared to the uncertainties introduced by the unknown size distribution below 0.1 um.

Line 189 to 190: Replace "likely" with "obviously". What other difference is there between the source and net fluxes than the deposition correction scheme?

Line 222: Please state the parameters used in the parametrization of Resio et al (2002).

Line 247 ". . . dependence of the sea spray flux . . ."

Line 265: May this be because above 2 um the deposition and settling velocities become so large that the correction term plays an too important role, thereby increasing the uncertainty of the source flux estimation?

Line 272 to 275: please specify to which size range the lognormal mode is fitted to? From the data there is no evidence that the peak of dF/dR80 is at R80=0.1um. I suggest that you also assume a lower mode center e.g. 50 nm or 70 nm, in order to estimate

the uncertainty of the correction for the missing film drop flux.

Line 276: What is the reason for integrating from 10 nm rather than from 1.5 nm?

Line 275 to 278: What is the reason for comparing the median of the correction to the mean of the number flux difference? Considering the large variability of the total number fluxes (Fig 3. B), the median of the number flux difference may be the more robust estimate.

---

## Author Comment (AC1) · 11 Nov 2019

Author comment 1 for "Sea Spray Fluxes from the Southwest Coast of the United

Kingdom –Dependence on Wind Speed and Wave Height" by Yang et al.

Many thanks for the constructive and supportive comments from reviewer #1, Chris Fairall, which we have kept in *Italic*. Please see our replies below.

*This paper is a description of an analysis of aerosol measurements made at Penlee Pt (PPAO) on the coast of the UK. Eddy correlation aerosol fluxes are computed and are interpreted as the sum of source and deposition fluxes. An effective aerosol source strength is computed and analyzed with considerations of wind direction, etc.. The authors find the source strength correlations better with wave height and/or wave parameters than with wind speed. The source is stronger than expected for open ocean but weaker than that observed directly from a surf zone. The interpretation is that wave breaking is more intense (or something) in a shallow zone close to shore than the open ocean. Most of the number flux occurs for aerosols greater than .1 and less than 1 micron. The paper is well written and the authors have carefully considered a number of experimental and physical aspects of the analysis and interpretation. In my view it can be published in its present form. I have a few comments the authors may with to consider.*

*\*I suggest the authors make it painfully clear that their results are not affected by surf generation and the enhanced production they see is associated with enhanced breaking in shallow water but external to the shore break. Maybe they thought it was obvious but I pondered this.*

Thanks for the suggestion. We did state this in the paragraph beginning at line 171 already but will make this point clearer. We will add in the abstract and the conclusion a sentence to the effect of "Sea spray fluxes measured from PPAO came essentially all from the shallow waters, and were not noticeably affected by the shore break/surf zone."

*\*I suggest they carefully check terminology of net, source, and total aerosols. I found myself wondering if they were consistent. A number of figures say 'total aerosol number flux' but I am not confident I know if it is net or source.*

Thanks for the suggestion. We will make it clearer in the revision that 'total' refers to either the CPC flux or the CLASP flux integrated over all size range; 'source' indicates total flux that has been corrected for deposition.

*Line 91. I don't think the turbulent flux is the same as the net flux. To me, net=turb-VgC, where Vg is the gravitational fall velocity.*
*Net=Source-VdC=turb-VgC*
*For the sizes they are considering, it may be that Vg is much less than Vd. They should state this. If Vg is negligible, then Source=turb+VdC*

Sorry for not explaining this more clearly in the original text. Our calculation of the aerosol deposition velocity accounts for size-dependent gravitational settling already. This will be made obvious in Section 2.

*On line 94 they state that source is obtained by subtracting deposition from net*
*Source=turb-VdC*
*Doesn't seem consistent with Fig 5, where source is greater than net. Please check this and get it straight.*

This is because we adopt the sign convention of Vd being negative. We will make this clearer in Section 2.

*Also suggest they read Andreas et al. JGR, vol 115, C12065, 2010; Freire et al BLM 160:249, 2016; and Nissanka et al. JGR, 9688:9702, 2018.*

Thanks for the suggested literature. They are consistent with our assessment that aerosol fluxes and concentrations are not in equilibrium at PPAO.

*Figure 11 is certainly interesting. It is surprising that aerosol spectral concentration and source flux is independent of wind speed. The graphs might be a little easier to use of the vertical axis was multiplied by R (are conserving).*

The aerosol source flux distribution is not quite independent of wind speed: fluxes of small aerosols are slightly higher at higher wind speeds (as indicated by the error bars), but the bin-separation is not nearly as clear as by wave height. We prefer to keep the vertical axes in dN/dR80 and dF/dR80 as they are the convention in literature.

---

## Author Comment (AC2) · 11 Nov 2019

Author comment 2 for "Sea Spray Fluxes from the Southwest Coast of the United

Kingdom –Dependence on Wind Speed and Wave Height" by Yang et al.

Many thanks for the detailed and supportive comments from reviewer #2, Sebastian Landwehr, which we have kept in *Italic*. Please see our replies below.

*This contribution describes total and size resolved Eddy Covariance measurements of aerosol number flux at a coastal research station. The total number flux measured with a CPC integrates particles larger than 1.5 nm. While size resolved fluxes between 0.1 and 6 um are derived from a CLASP. The authors combine the measured net fluxes with size resolved dry deposition velocities derived from the model of Slinn and Slinn (1980), in order to obtain the source flux over the footprint area. The dependence of the size resolved and size-integrated source flux on wind speed, wave height, and wave Reynolds number is studied in detail and compared with previous parametrizations that where derived over the open ocean and in coastal waters. The authors conclude that their measurements and findings are representative for coastal waters (excluding the surf zone) where sea spray emissions are largely enhanced due to costal wave breaking. Eddy Covariance measurements of aerosol fluxes in the marine environment provide an almost direct way to measure sea spray emission fluxes, however due to the complexity of making these measurements only few data sets exist today. Quantification of coastal sea spray emissions are relevant to the aerosol concentration and composition in the densely populated coastal regions and may thus bear relevance for heath related studies. Comparing the relation of coastal and open ocean sea spray emissions to forcing parameters may help to further improve the understanding of sea spray emission in general. I therefore consider this paper to be a valuable contribution and recommend publication with minor revisions as outlined below.*

*\*As stated by referee Christopher Fairall the source flux can be derived from the measured turbulent flux (FEC) via Source = FEC+Vd\*C-Vg\*C, where C is the particle concentration, and Vd and Vg are the dry deposition and the settling velocity respectively. Vg depends on the particle size, shape and density, and hence on the relative humidity and the hygroscopy of the particles.*

Our calculation of the aerosol deposition velocity accounts for size-dependent gravitational settling already. We apologize for not making this clearer in the original paper.

*The deposition velocity Vd may in additional be affected by turbulent and diffusive processes. Various dry deposition parametrizations can be found in the literature. The authors settle for the model of Slinn and Slinn (1980), which assumes that the particles assume their dry diameter in the constant flux layer and grow to a diameter defined by the growth rate and the assumed equilibrium relative humidity above seawater. Depending on the ambient relative humidity, the assumption of dry sea salt particles in the constant flux layer may introduce some errors. I therefore recommend the authors to*

*explicitly state how they apply Slinn and Slinn (1980), i.e. which diameter conversions are performed, what particle densities are assumed.*

We compute the aerosol deposition velocity (accounting for gravitational settling) per Eq. 4 from Slinn and Slinn 1980.
Aerosol size change as a function of humidity is computed following Gerber (1985) and we assume a dry aerosol density of 2170 kg/m3 (that of sea salt).

*\*I further suggest that the ambient relative humidity time series should be presented in Fig 3., since it should play an important role in determining the correction term (Vd\*C-Vg\*C).*

A time series of relative humidity during this example period is shown below.

[Figure]

*Line 103 to 105: Please consider splitting this sentence into two. Maybe the time series of the correction factors could be provided in the appendix to illustrate the variability of the relative importance of the correction term.*
We have revised the sentence to: *"Integrated over all CLASP sizes, this deposition correction amounts to 25 cm$^{-2}$ s$^{-1}$ in the mean (up to ~200 cm$^{-2}$ s$^{-1}$) when winds were from the southwest.  This represents 14% of the net flux on average (up to ~50%)."* The total deposition correction term for the CLASP (including gravitational settling) is shown in the figure above, and we adopt the formula of source flux = net flux – deposition_correction.

*Line 109: Please specify which of the formulations, provided in Gerber (1985), you are using and with which set of coefficients?*

The formulations and coefficients (in Matlab) are copied below.
A=0.7674; B=3.079; C=2.572E-11; D= -1.424; % seasalt coefficients

% define dry radius space
Rd = [0.0001:0.0001:20]';

 % grow dry radii to ambient humidity RH1 (RH1 between 0 and 1)
 Ra = ( (A*Rd.^B)./(C*Rd.^D - 0.434*log(RH1)) + Rd.^3 ).^(1/3);

```
% find dry radii for radii measured at RH1
 for n=1:length(R1);
   k = max(find(Ra<R1(n)));
   R1d(n) = Rd(k)+diff(Rd(k:k+1))*((R1(n)-Ra(k))/diff(Ra(k:k+1)));
 end
```

% and calculate the radii grown back to required humidity RH2 (here 0.8, or 80%)
R2 = ( (A*R1d.^B)./(C*R1d.^D - 0.434*log(RH2)) + R1d.^3 ).^(1/3);

*Line 115: please correct "off of"*
We have replaced 'off of' with 'from.'

*Line 138: R (=radius) has not been defined previously. Theres is some chance of confusion with the coefficient of determination. At this note, please unify the use of italic and noraml font for variables like Hs and RHw.*
We have chosen to spell out radius here and not use R to indicate radius. We have made sure that significant wave height and wave Reynolds number are thoroughly italicized in the revision.

*Line 134 to 142: How much does Vd predicted by Slinn and Slinn (1980) vary over the range of conditions encountered during the observations? Would accounting for these variations reduce the scatter in Figures 4, 6, and 7? However, these variations may be small compared to the uncertainties introduced by the unknown size distribution below 0.1 um.*
Vd varied from 0.007 to 0.13 cm/s during this period. The wind dependent Vd correction WAS applied to the CPC data, thus accounting for these variations already. We have revised the sentence to the following to be clearer: 'We compute the wind speed dependent $V_d$ at these two aerosol sizes using the Slinn and Slinn (1980) parameterization, which amounts to 0.034 and 0.010 cm s$^{-1}$ for the mean conditions at PPAO. For simplicity, we take the average of the two $V_d$ datasets and multiply it by the CPC number concentration to estimate the deposition flux.'

*Line 189 to 190: Replace "likely" with "obviously". What other difference is there between the source and net fluxes than the deposition correction scheme?*
We have changed the sentence to "The source fluxes show greater discrepancy due to the different deposition correction schemes applied."

*Line 222: Please state the parameters used in the parametrization of Resio et al (2002).*
We have revised the line to "Here $H_s$ in the Plymouth Sound is predicted using a parameterization for fetch-limited waters (Resio et al. 2002) as a function of fetch and friction velocity since a direct measurement is not available."

*Line 247 ". . . dependence of the sea spray flux . . ."*
Suggestion accepted.

*Line 265: May this be because above 2 um the deposition and settling velocities become so large that the correction term plays an too important role, thereby increasing the uncertainty of the source flux estimation?*
Our statement was on the number distribution, and not on the fluxes.

*Line 272 to 275: please specify to which size range the lognormal mode is fitted to? From the data there is no evidence that the peak of dF/dR80 is at R80=0.1um. I suggest that you also assume a lower mode center e.g. 50 nm or 70 nm, in order to estimate he uncertainty of the correction for the missing film drop flux.*
The lognormal mode was allowed to be fitted from 10 nm to 7 micron radius, but in practice has a width of approximately 80 to 150 nm in our fit.
We refrain from making a lognormal fit centered at a lower mode (e.g. 50 or 70 nm) for two reasons: first, with less than half of the mode captured by the CLASP, the uncertainty of the fit would be very large; second, previous measurements of marine aerosols often show a mode at a measured radius of 100 nm, hence our assumption that the lowest bin measured by the CLASP represents this mode size. Future measurements of finer aerosol size distributions (e.g. by a SMPS) at our site should provide more information on the robustness of our assumption.

*Line 276: What is the reason for integrating from 10 nm rather than from 1.5 nm?*
It makes no difference whether the log-normal fit is integrated from 10 nm or from 1.5 nm, as the fit is already at zero at 10 nm.

*Line 275 to 278: What is the reason for comparing the median of the correction to the mean of the number flux difference? Considering the large variability of the total number fluxes (Fig 3. B), the median of the number flux difference may be the more robust estimate.*
Good point. We have revised the sentence to "This is consistent with the finding that the total aerosol number flux from the CLASP was lower than that from the CPC by a mean (median) of 45 (40) cm$^{-2}$ s$^{-1}$."